# *Pseudomonas* PS01 Isolated from Maize Rhizosphere Alters Root System Architecture and Promotes Plant Growth

**DOI:** 10.3390/microorganisms8040471

**Published:** 2020-03-26

**Authors:** Thanh Nguyen Chu, Le Van Bui, Minh Thi Thanh Hoang

**Affiliations:** 1Faculty of Biology and Biotechnology, University of Science-Ho Chi Minh City, Ho Chi Minh City 700000, Vietnam; bvanle@hcmus.edu.vn; 2Laboratory of Molecular Biotechnology, Vietnam National University, Ho Chi Minh City 700000, Vietnam

**Keywords:** indole-3-acetic acid (IAA), plant growth-promoting rhizobacteria (PGPR), *Pseudomonas*, root system architecture (RSA), volatile organic compounds (VOCs), whole-genome sequencing

## Abstract

The objectives of this study were to evaluate the plant growth promoting effects on *Arabidopsis* by *Pseudomonas* sp. strains associated with rhizosphere of crop plants grown in Mekong Delta, Vietnam. Out of all the screened isolates, *Pseudomonas* PS01 isolated from maize rhizosphere showed the most prominent plant growth promoting effects on *Arabidopsis* and maize (*Zea mays*). We also found that PS01 altered root system architecture (RSA). The full genome of PS01 was resolved using high-throughput sequencing. Phylogenetic analysis identified PS01 as a member of the *Pseudomonas putida* subclade, which is closely related to *Pseudomonas taiwanensis*. PS01 genome size is 5.3 Mb, assembled in 71 scaffolds comprising of 4820 putative coding sequence. PS01 encodes genes for the indole-3-acetic acid (IAA), acetoin and 2,3-butanediol biosynthesis pathways. PS01 promoted the growth of *Arabidopsis* and altered the root system architecture by inhibiting primary root elongation and promoting lateral root and root hair formation. By employing gene expression analysis, genetic screening and pharmacological approaches, we suggested that the plant-growth promoting effects of PS01 and the alteration of RSA might be independent of bacterial auxin and could be caused by a combination of different diffusible compounds and volatile organic compounds (VOCs). Taken together, our results suggest that PS01 is a potential candidate to be used as bio-fertilizer agent for enhancing plant growth.

## 1. Introduction

Plant growth-promoting rhizobacteria (PGPR) colonize the rhizosphere and play a key role in the improvement of plant fitness and yield [1]. The genus *Pseudomonas* is one of the most abundant genera of PGPR, which is capable of enhancing plant growth and modulating the root system architecture in different plant species [2,3,4]. For examples, *Pseudomonas nitroreducens* enhances the growth of *Arabidopsis thaliana* and *Lactuca sativa* [5]; *Pseudomonas putida*, *Pseudomonas aeruginosa* and *Pseudomonas fluorescens* stimulate growth of various crop plants such as maize (*Zea mays*), soybean (*Glycine max*), wheat (*Triticum*), peanut (*Arachis hypogaea*) and mung bean (*Vigna radiata*) [6,7,8,9,10,11]. In addition, *Pseudomonas* spp. strains including *Pseudomonas simiae* WCS417 (formerly *P. fluorescens* WCS417), *P. fluorescens* WCS347 and *P. putida* WCS358 also promote plant growth and drive root plasticity in *Arabidopsis* by inhibiting primary root elongation and promoting lateral root development and root hair formation [3].

*Pseudomonas* spp. promote plant growth or modulate root development by producing and secreting phytohormones (e.g., indole-3-acetic acid, IAA) and/or volatile organic compounds (VOCs) [4,12,13,14,15,16,17]. Some Pseudomonas species produce IAA, which can enhance plant growth and alter root system architecture. The importance of auxin in the root architectural alteration was demonstrated by inoculation of *P. aeruginosa*, which is accompanied by increased auxin-responsive gene expression [18]. Likewise, *P. aeruginosa, P. putida* and *P. fluorescens* can modulate root system architecture through an auxin signaling response mediated by cyclodipeptides [4,18]. Bacterial IAA synthesis mainly depends on tryptophan (Trp) via five distinct pathways including indole-3-acetamide (IAM), indole-3-acetonitrile (IAN), indole-3-pyruvic acid (IPyA), tryptamine (TAM) and tryptophan side-chain oxidase (TSO) pathways [19,20,21]. Some bacterial VOCs such as 2,3-butanediol, 3-hydroxy-2- butanone, 2-pentylfuran, *N*,*N*-dimethyl-hexadecanamine, carbon dioxide (CO_2_), 13-tetradecadien-1-ol, 2-butanone and 2-methyl-n-1- tridecene promote plant growth and modulate root system architecture. For instance, VOCs produced by *P. simiae* WCS417 stimulate lateral root formation [3]. Moreover, *P. fluorescens* SS101 promotes the growth of plants via the release of VOCs including 13-Tetradecadien- 1-ol, 2-butanone and 2-Methyl-n-1-tridecene [17].

Regarding the potential health risk of PGPR to human, efforts in characterization of the isolates at the species level should be employed before proceeding with formulation development or pilot and largescale field level applications. Currently, agricultural microbiologists are mainly focused on the beneficial traits of the isolates while ignoring the possible existence of opportunistic human or animal pathogenicity [22,23]. Therefore, it is highly desirable to use phylogenomic data for exact taxonomy classification. In this context, high-throughput sequencing technologies and genome annotation have provided whole-genome sequencing as an efficient tool for not only bacteria classification but also provide insight into the molecular mechanisms and functional capabilities of PGPR [24,25]. Recently, whole genome sequencing has been applied to study the genome of some Pseudomonas species, such as, Pseudomonas sp. UW4 and Pseudomonas *chlororaphis* strains, HT66, GP72, 30–84 and O6; *Pseudomonas* WCS358, WCS374 and WCS417, to identify genes contributing to plant growth-promoting activities [26,27,28].

The Mekong Delta in Vietnam is the largest agricultural region of the nation, which make it highly vulnerable to climate change [29]. It is also fighting soil degradation due to poor agricultural practices, with primary example being the use of uncontrolled chemical fertilizers. Application of PGPR as bio-inoculants, as a replacement for chemical fertilizer, therefore, is considered a sustainable agricultural practice. Moreover, it is a promising strategy for environmental safety and food security in response to climate changes [30,31]. It is important to isolate and identify region-specific microbial strains from specific crop plants, which can be used as the potential plant growth promoters to achieve desired crop production [30,32,33]. The isolation and screening of PGPRs from the Mekong Delta, Vietnam was reported previously [34,35]. However, the molecular mechanisms for plant growth enhancement and the genome sequences of native PGPRs have not been described yet. Thereby, the present study aims to (i) screen and select a novel strain of *Pseudomonas* spp. which was isolated from rhizospheres of some plant species grown on the Mekong Delta, (ii) characterize the effects of the isolate with respect to plant growth promotion and alteration of root system architecture (RSA), and (iii) investigate the relevant molecular mechanisms. The complete genome of a novel isolate, PS01 strain, was sequenced. We identified several gene clusters that likely contribute to plant growth-promoting properties of PS01. To further investigate mechanisms of PS01- plants interaction, we also performed the transcriptional expression, genetic and pharmacological interference of auxin inhibitors analysis using *Arabidopsis* as a model plant. Our results suggested that *Pseudomonas* PS01-induced modulation of the RSA and plant growth probably involves distinct signaling pathways, and that *Pseudomonas* PS01 could be commercialized as a bio-inoculant, which could be applied for a sustainable agriculture development.

## 2. Materials and Methods

### 2.1. Source of Plant Growth-Promoting Rhizobacteria and Bacterial Growth Condition

Twenty *Pseudomonas* isolates from various plant rhizospheres (Table 1) collected from the Mekong Delta were isolated on King’s B medium. All bacterial strains were stored at −80 °C in Tryptone Soya Broth (TSB) supplemented with 10% glycerol. For particular experiments, a stock culture of each strain was freshly inoculated in 10 mL of TSB and was grown for 18 h at 30 °C / 120 rpm.

### 2.2. Plant Growth Promoting Characterization of Bacterial Strains

Nitrogen-fixing ability of the pure bacterial culture was determined on N-free medium as described by Bashan et al. [36]. The ability of the isolate to solubilize the inorganic tricalcium phosphate [Ca_3_(PO_4_)_2_] was checked on Pikovskaya’s agar [37]. IAA production ability was identified using Salkowaski’s reagent [38].

### 2.3. Plant Material and Growth Conditions

Seeds of *Arabidopsis thaliana* wild type Col-0 and mutant lines including *eir*1-1; *axr*4-1; *aux*1-7 as well as *DR5::GUS* transgenic line were kindly provided by Dr. Philippe Nacry (Biochemistry and Plant Molecular Physiology, Montpellier, France). Seeds were surface-sterilized with 5% sodium hypochlorite (*v*/*v*) supplemented with 0.1% Triton × 100 (*v*/*v*) for 5 min and washed five times with sterile distilled water. After stratification at 4 °C for 2–3 days, seeds were sown on half-strength Murashige and Skoog (MS ½) solid medium supplemented with 1% sucrose. Plates were kept vertically in the growth chamber at 22 °C with a photoperiod of 16 h light and 8 h dark. Four days after germination, uniform seedlings with similar size were transferred onto new (MS ½) solid medium plates as described previously [5].

### 2.4. Screening Pseudomonas Strain for Plant Growth Promotion on Arabidopsis Thaliana (Col-0)

Four-day-old seedlings were transferred either to new solid MS ½ medium (as a control) or to inoculation medium, in which, the bacteria inoculum was placed at 2.5 cm below the root tips. Twelve days after inoculation, the shoots and roots were separately harvested. Media adhering to the roots were carefully removed using forceps. Then the roots were thoroughly rinsed with distilled water and blotted dry prior to fresh biomass determination. The effects of *Pseudomonas* isolates on plant growth were evaluated by comparing shoot and root fresh weight of inoculated plants to that of the control plants. The *Pseudomonas* isolate showing the highest plant growth promotion was selected for further experiments.

### 2.5. Root Architecture Analysis of Arabidopsis Thaliana

To investigate the effect of bacteria on root development, the 4 days-old seedlings were inoculated with bacteria as described in Section 2.4. Seven days after inoculation, seedling roots were photographed and analyzed using the ImageJ software (available at http://rsb.info.nih.gov/ij/; developed by Wayne Rasband, National Institutes of Health, Bethesda, MD, USA). The number of emerged lateral roots (LRs) was counted using a microscope. The number of root hairs (RHs) from primary root segments located 1 mm above the first RH was counted using a microscope at 100 × magnification. RH length was analyzed using the ImageJ software.

### 2.6. Identification of Bacterial Factors for Root System Alteration by PS01

To define whether the diffusible compounds produced by PS01 had an effect on the root development, the 4 days-old seedlings were treated with cell-free culture supernatant of PS01 growing on TSB medium supplemented with or without tryptophan. For bacterial supernatant preparation, PS01 was enriched in TSB or TSB supplemented with 1% Trp (w/v) for 72 h in a 30 °C. Cultures were centrifuged (5000× *g*, 10 min) and the resulting supernatant was sterilized by filtration. To assess the effect of VOCs produced by PS01 on root system alteration, 20 µL of bacterial suspension (10^6^ CFU/mL) was applied in a paper disk which was placed in one part of the split plate [3,39]. The root system architecture was analyzed after 7 days of inoculation.

For inhibition of auxin reception, the 4 days-old seedlings were transferred and grown vertically on MS ½ media supplemented 5 µM auxinole with or without PS01 [40]. For auxin transport inhibitor, naphthyl phthalamic acid (NPA) was added at a final concentration of 1 or 5 µM [3]. The root analyses were assessed after 9 days of inoculation.

### 2.7. GUS Histochemical Assay

The effect of bacteria inoculation on *DR5* gene expresson in *A. thaliana* seedlings was analyzed using transgenic DR5::GUS plants. The whole seedlings were incubated in GUS solution (50 mM sodium phosphate [pH 7], 10 mM EDTA (Ethylene Diamine Tetraacetic Acid), 0.5 mM K_4_[Fe(CN)_6_], 0.5 mM K_3_[Fe(CN)_6_], 0.5 mM 5-bromo-4-chloro-3-indolyl-b-glucuronic acid, and 0.01% Silwet L-77) at 37 °C for 2 h [41].

### 2.8. Genome Sequencing and Annotation

Bacterial genomic DNA was extracted and processed according to the Nextera XT library preparation kit (USA), and sequenced using Illumina NextSeq500 instrument. Library preparation and sequencing were performed at the Research Resources Center, University of Illinois at Chicago. Reads were trimmed and filtered by Trimmomatic v0.36 (written by Anthony Bolger from the Bjorn Usadel Lab) with FastQC v0.11.77 (https://www.bioinformatics.babraham.ac.uk/projects/fastqc). De novo assembly was performed by using Velvet 1.2.09 (developed by Daniel Zerbino and Ewan Birney, European Bioinformatics Institute, United Kingdom) with K-mer of 49. The statistics of contigs were shown by using QUAST web server (http://quast.sourceforge.net) and the plot of coverage shown by Artermis 16.0.0. The gene detection and genome annotation were performed using the RAST (http://rast.nmpdr.org/) server and BlastKOALA v2.1 (http://www.kegg.jp/blastkoala/). Later, the annotated genes relating in plant growth-promoting pathways were investigated. Phylogenetic affiliation of bacteria was assessed using Insert Genome Into Species Tree 2.1.10 tool (http://kbase.us). At the same time, Ribosomal Multilocus Sequence Typing (rMLST) was performed for identification of the phylogenetic position of bacteria (https://pubmlst.org/rmlst). This approach is based on indexes variation of the 53 genes encoding the bacterial ribosome protein subunits (*rps* genes) [42].

### 2.9. RNA Extraction and Real-time Quantitative-PCR (RT-qPCR) Analyses

RNA was extracted from 24 h and 48 h treated and non-treated seedlings. Roots and shoots from at least 50 seedlings were harvested separately and RNA was extracted using Trizol (Invitrogen™, USA) method. RT-qPCR was performed using the Luna Universal One-Step RT-PCR Kit (New England Biolabs, USA). Real-time qPCR was performed on the Light Cycler 96 System (Roche). The reactions were incubated at 50 °C for 2 min and 95 °C for 10 min, followed by 40 cycles of 95 °C for 15 s and 60 °C for 1 min. The specificity of the real-time PCR amplification products was checked with the following dissociation protocol: heating at 95 °C for 15 s, cooling at 60 °C for 20 s, slowly heating up to 95 °C within 20 min and a heating plateau at 95 °C for 15 s. Specific primer sets (Appendix A) were reported previously [43]. To confirm the specificity, the size of PCR product was estimated by gel electrophoresis. The relative transcript level (RTL) was calculated by normalizing to *ACT2* expression level as follows: RTL  =  2∆∆Ct, where ∆∆Ct  =  ∆Ct (gene) − ∆Ct (*ACT2*). Experiments were performed with three biological and two technical replicates.

### 2.10. Pot Experiment

Corn seeds (TrangNong Company, Vietnam) were surface-sterilized as mentioned above. Sterilized seeds were coated with bacteria by dipping in bacterial suspension (10^8^ CFU/mL) for 1 h. For control treatment, the seeds were dipped in sterile distilled water. Then seeds were germinated on sterilized wet cotton in Petri dishes at 30 °C. After 2 days of incubation, the germination rate of seeds was recorded. Germinated seeds were sown in clay pots (1 seeds per pot) containing 0.5 dm^3^ sterilized soil. The pots were kept in green house condition (28–35 °C/65–80% RH), administered with tap water daily (100 mL per pot). Bacteria suspension (1 mL of 10^8^ CFU/mL) was mixed into water (100 mL per pot) and irrigated once a week. TSB was used for the control seedlings. Chemical fertilizer was not applied to the plants during the experiment. Thirty days after sowing, the photosystem II efficiency in a dark-adapted state (Fv/Fm) was measured using FlourPen FP 100. Soil adhering to the roots was carefully removed by washing with tap water. The shoots and roots were separately harvested and dried in an oven for 2 days at 70 °C. Then the shoot and root dried-weight of plants were recorded (*n* = 15).

In order to test root colonization by bacteria, soil adhering to the roots were carefully removed and 10 g of root fresh weight per plant was suspended in phosphate-buffered saline. Then, serial dilutions were plated on King’s B medium and were incubated at 30 °C for 48 h. The colony-forming units (CFU) per gram of fresh root were determined as previously described [44].

### 2.11. Statistical Analysis

Statistical data were computed using the GraphPad Prism 7 program. Data were first tested for normality with the D’Agostino & Pearson omnibus normality test and for homogeneity of variance with the Brown–Forsythe test. ANOVA followed by Tukey’s Honestly Significant Difference (HSD) test was performed for comparisons among all means and Student’s *t* test was performed for comparison of two means. Error bars showed standard deviation and asterisks indicate statistical significance of the differences (*p* < 0.05).

## 3. Results

### 3.1. Plant Growth Promoting Characterization of Bacteria.

In the previous studies, we found that *Pseudomonas* strain PS01 could enhance plant growth in saline conditions, whereas *Pseudomonas* strains P112 and P113 (coded as PS02 and PS03 in this study) enhanced plant growth in normal conditions [35,45]. In this study, we evaluate plant growth-promoting (PGP) traits of these strains and 17 other strains which were isolated from the rhizosphere and rhizoplane of various crop plants. The PGP traits including IAA production, phosphate solubilization and N fixation are listed in Table 1. Four out of 20 isolates were able to fix nitrogen. All the isolates were able to produce IAA with a range of 0.42–46.75 μg ml^−1^ in the presence of IAA precursor tryptophan after 72 h of incubation. All isolates formed a clear zone of calcium phosphate solubilisation on Pikovskaya’s agar plates after seven days of incubation.

**Table 1 microorganisms-08-00471-t001:** Plant growth promoting characters of bacteria strains.

Code of Strains	Host Plant	Conc. IAA (ug/mL) ± SD	Phosphate Solubilization Index (*)	Nitrogen Fixation Ability	Ref.
PS01	*Zea mays*	8.94 ± 0.29	1.34	-	[45]
PS02	*Zea mays*	46.75 ± 3.87	1.92	+	[35]
PS03	*Zea mays*	34.94 ± 4.46	1.76	+	[35]
PS04	*Zea mays*	25.50 ± 0.8	1.85	-	This study
PS05	*Killinga nemoralis*	32.22 ± 0.86	1.82	-	This study
PS06	*Killinga nemoralis*	17.36 ± 0.21	1.36	-	This study
PS07	*Musa*	0.85 ± 0.23	1.40	+	This study
PS08	*Musa*	35.62 ± 1.14	1.35	-	This study
PS09	*Eleusine indica*	22.19 ± 0.69	1.66	-	This study
PS10	*Eleusine indica*	0.71 ± 0.13	1.73	-	This study
PS11	*Saccharum* L.	20.95 ± 1.58	1.51	-	This study
PS12	*Mentha aquatic*	0.42 ± 0.25	1.39	-	This study
PS13	*Mentha aquatic*	26.64 ± 3.11	1.63	-	This study
PS14	*Coix lacryma-jobi*	18.68 ± 0.49	1.73	-	This study
PS15	*Phyllanthus amarus*	6.36 ± 0.1	1.45	-	This study
PS16	*Piper lolot*	13.35 ± 0.58	1.34	+	This study
PS17	*Oryza* sp.	6.12 ± 0.68	1.74	-	This study
PS18	*Ipomoea aquatica*	16.09 ± 0.83	1.34	-	This study
PS19	*Amaranthus*	1.65 ± 0.5	1.45	-	This study
PS20	*Brassica integrifolia*	6.60 ± 0.15	1.48	-	This study

(*) Phosphate solubilization index was derived by dividing the total diameter of the clear zone (colony+clear zone) with diameter of the colony. IAA: Indole-3-acetic acid.

### 3.2. Screening for Effective PGPR Strains Using Plant Growth Assay

To investigate the plant growth-promoting effects, all bacterial isolates were evaluated using *A*. *thaliana* ecotype Columbia (Col-0). After 12 days of co-cultivation, the shoot and root fresh weight of seedlings were measured. The inoculation of isolated bacterial strains on *A. thaliana* showed a wide range of effects from promoting to suppressing plant growth (Figure 1). Twelve isolates significantly increased the plant biomass. The root and shoot fresh weight of seedlings inoculated with PS01 showed the maximum increases of 4.8 and 2.4 fold, respectively, compared to non-inoculated seedlings (Figure 1). In contrast, a slight decrease in root and shoot fresh weight was observed upon inoculation with two isolates (PS06 and PS12). On average, PS01 strain showed most prominent plant growth promoting effects. Thus, PS01 strain was selected to further evaluate its effect on the responses of *Arabidopsis* root system.

### 3.3. Effects of Pseudomonas PS01 Strain on RSA of Arabidopsis Thaliana

We used several parameters to represent RSA, including primary root lengths, the number of lateral roots and root hairs as well as root hairs lengths. After 7 days of co-cultivation, the total root length increased 1.3 fold in seedlings inoculated with PS01 compared to control plants (Figure 2a). Interestingly, the primary root length of these plants was reduced by approximately 1.7 fold (Figure 2b) while LR density increased 4.1 fold in inoculated seedlings (Figure 2c). Furthermore, the second order lateral roots were observed only in seedlings inoculated with PS01. The root hair formation sites in the inoculated roots were located closer to the root tip than that in the non-inoculated roots (Figure 2f). The lengths of non-root hair-producing epidermal cells in the differentiation zone of inoculated primary roots were shorter than those in non-inoculated roots (Appendix A). This suggests that the reduction of primary root length by PS01 was related to the inhibition of cell expansion in the elongation zone. The kinetic of LR initiation in response to PS01 was assessed over a 5-days period using microscopy in order to observe LR initiation sites across the primary roots (Appendix A). Our data showed LR initiation occurred only after 2 days of co-cultivation in PS01-inoculated seedlings compared to 3 days in control roots. Moreover, PS01 also promoted RH development. In particular, the RH density and the average RH length increased 4 fold and 2.5 fold, respectively, compared to the non-inoculated roots (Figure 2d,e). Collectively, the data indicates that PS01 inhibited primary root elongation, while promoting lateral root formation and overall root hair development.

### 3.4. Genome Characteristics and Phylogenetic Analysis of Pseudomonas PS01

The genus *Pseudomonas* is very diverse and is still undergoing taxonomic refinement [46]. Previously, PS01 was identified as a member of the *Pseudomonas putida* group based on *rpoD* gene sequence [45]. In this study, to further refine the taxonomic classification of PS01, the whole genome of this strain had been sequenced. A total of 3916,338 raw reads (251 bp in length) producing approximately 30-fold coverage of the PS01 genome were generated. These raw reads were trimmed and filtered with a low-quality average (≤Q20) resulting in 3063,956 clean reads which were then used for *de novo* assembly. Finally, we generated 71 contigs with a total length of 5338,160 bp using k-mer 49. The largest contig is 470,750 bp in length (N50 = 92905 bp). PS01 genome contains 4820 putative coding sequences (CDS), number of RNAs is 55 and the overall guanine-cytosine (GC) content is about 62.07%. The phylogenetic tree suggested that PS01 is a member of *Pseudomonas putida* subclade and the closest relative is *P. taiwanensis* (Figure 3). rMLST tool also showed the similar result (Support 100%).

### 3.5. Identification of Genes Associated with IAA and VOCs Production in Genome of Pseudomonas PS01 Strain

Bacterial IAA production is reported as a major factor in plant growth promotion as well as RSA alteration [47,48]. Therefore, we tested the presence of IAA biosynthetic pathways in PS01 strain. Interestingly, we identified the complete IAM and TAM pathways for IAA biosynthesis in PS01 genome (Appendix A). These pathways were also reported in *P. putida W619* while TSO and IAM pathways were identified in *P. fluorescens Psd* [49,50]. In *P. chlororaphis* O6, only TAM pathway was reported [51]. Experimental results showed that PS01 strain can synthesize IAA in TSB medium supplemented with Trp (Table 1). Surprisingly, IAA content produced by PS01 strain (8.94 µg/mL) was much lower than that produced by PS02 (46.75 µg/mL) or PS08 strains (35.62 µg/mL) (Table 1). This suggests that IAA production may not be the sole factor for the plant growth promoting activity of PS01.

In addition to IAA, VOCs produced by bacteria may not only promote plant growth by modifying the root architecture system but also by playing an important role in inducing systemic resistance against plant pathogen. It has been suggested that 2,3-butanediol and its precursor acetoin play a role on plant growth promotion [52]. In this study, genes involved in acetoin and 2,3-butaneldiol synthesis such as acetolactate synthase, acetolactate decarboxylase, butaneldiol dehydrogenase and acetoin reductase were identified in the PS01 genome (Appendix A). Furthermore, the *ubiC* gene, which is implicated in 4-hydroxybenzoate biosynthesis, was also present in PS01 genome. 4-hydroxybenzoate is a VOC that can act as an antibiotic against plant pathogens. Its biosynthesis pathway was found in the genome of several PGPRs, such as *Pseudomonas protegens* Pf-5 or *Pseudomonas* sp. UW 4 [26]. Interestingly, the *ubiC* gene has been identified in all 21 *Pseudomonas* species, therefore, it has been suggested that the 4-hydroxybenzoate biosynthesis is a common pathway in *Pseudomonas* spp. [26].

### 3.6. Determination of Bacterial Factors Involved in Alteration of RSA

To identify bacterial determinants involved in RSA modulation, we analyzed the role of diffusible metabolites and VOCs produced by PS01. The results showed that primary root length was reduced approximately 50% and 40% in seedlings treated with cell-free culture supernatant of PS01 grown in medium with or without Trp (precursor of IAA), respectively, compared to the control plants (Figure 4a). In contrast, primary roots of seedlings exposed to bacterial VOCs were not inhibited (Figure 4a). Moreover, the LR number increased in seedlings exposed to bacterial VOCs (Figure 4b). The average number of LRs in treated seedlings was 7.3, while in control seedlings is 1.9 (TSB with Trp) and 2.0 (TSB without Trp. Plants treated with cell-free cultures showed no statistically significant increase in their number of LRs. On the other hand, the number and length of RHs increased significantly in seedlings treated with cell-free cultures and bacterial VOCs (Figure 4c,d). Interestingly, VOCs from PS01 promotes LR initiation while not affecting primary root growth. By contrast, cell-free culture of PS01 stimulated RH development but did not affect LR formation.

Taken together, our data suggest that PS01 diffusible compounds and VOCs played a major role in elongation of primary roots and induction of LRs initiation, respectively. Furthermore, the RHs promotion by PS01 is related to the combination of diffusible unknown compounds and VOCs. However, the contribution of each component to RHs promotion effect has not been evaluated. Taken together, the RSA responses triggered by PS01, including primary root growth, LRs formation and RHs development, in *Arabidopsis thaliana* may involve multiple secreted metabolites.

### 3.7. Role of Bacterial IAA on RSA

As described above, analysis of the PS01 genome showed the presence of two different biosynthetic routes of bacterial IAA production, IAM and TAM pathways. However, the IAA amount synthetized by PS01 strain was relatively low compared with other *Pseusomonas* strains in culture media supplemented with Trp. Even IAA was not detected in culture media without Trp. These results raise the question of whether the amount of IAA produced by PS01 strain is sufficient to alter the RSA in *Arabidopsis* co-cultivation experiments.

To further address the effect of IAA produced by PS01 on the RSA, the SCF^TIR1/AFB^ (SKP-Cullin-F box (SCF), Transport Inhibitor Resistant1/Auxin signaling F-box (TIR1/AFB)) auxin receptor inhibitor, auxinole (α-[2,4-dimethylphenylethyl-2-oxo]-IAA) was supplemented to the medium to block the auxin perception [40,53]. The results showed that the LR formation was completely blocked in the presence of 5µM auxinole (Figure 5). In the presence of auxinole, the number of emerged LRs in PS01-inoculated seedlings (3.8 per plant) was significantly higher compared to that of non-inoculated seedlings (not detected). However, the ability of PS01 to promote LR initiation was severely decreased in the presence of auxinole (Figure 5a). These results correspond the results of split plate assay (Figure 4), suggesting that bacterial IAA (a diffusible compound) is not directly related to promotion of LR initiation by PS01, but a functional auxin perception machinery in plants are required. Moreover, the RH formation induced by PS01, was only slightly affected (Figure 5b) while the length of RH was unaffected (Figure 5c). PS01-triggered promotion of RH development was further evaluated in the *Arabidopsis* auxin-resistant mutant *axr*4-1. The *axr*4 auxin-resistant mutants are specifically resistant to exogenous auxin [54]. The results showed that the ability of PS01 to promote RH development remained almost unalterable in the mutant *axr*4-1. (Figure 5d,e).

As described above, the split plate assay showed that primary root length did not change in roots exposed to the volatile blend of PS01 while significantly decreasing in roots inoculated with PS01 as well as roots treated with cell-free culture. This suggests that the inhibition effect on inoculated primary root is not due to bacterial VOCs but, rather, to unknown bacterial diffusible compounds. The typical response of roots to exogenous auxin is limited elongation; therefore, using auxinole induced a slight increase in primary root length in non-inoculated seedlings (Data not shown). However, limited primary root growth remained in the inoculated seedlings suggested that bacterial IAA was not directly involved.

To determine if PS01 induces LR formation via bacterial IAA-dependent pathway, the transcript levels of genes involved in auxin-dependent lateral root development regulation (*AIR12*, *DFL1* and *NAC1*) (Appendix A) [43] was analyzed after 24 h and 48 h of incubation with PS01 (Figure 6). We found that *AIR12* expression in roots was up-regulated after 48 h inoculation. Bacterial inoculation did not induce any changes in the transcriptional expression of the *DFL1* and *NAC1* in either the root or the shoot.

### 3.8. Effect of PS01 on Endogenous Auxin Biosynthetic and Distribution of Auxin in Roots

Next, to investigate whether PS01 alters endogenous auxin levels in plants, transcriptional levels of genes coding for enzymes involved in tryptophan and auxin biosynthesis (*AAO1*, *ASA1*, *TSA1*, *YUCCA5*, *CYP79 B2*, *CYP83 B1* and *IGPS*) [43] were analyzed. In both shoots and roots, the relative transcriptional levels of the *AAO1*, *ASA1*, *TSA1*, *YUCCA5*, *CYP79 B2*, *IGPS* showed no changes after inoculation with PS01. The expression level of *CYP83 B1* was slightly down-regulated in the roots after 24 h of co-cultivation while it was not significantly affected after 48 h of co-cultivation. Inoculation with PS01 down-regulated *CYP83 B1* expression in shoot after 24 h of co-cultivation but induced an increase in its transcript abundance after 48 h of co-cultivation (Figure 7). Therefore, the transcriptional data suggested that PS01 might slightly stimulate auxin biosynthesis in *Arabidopsis* roots after 24 h.

In addition to genes involved in tryptophan and auxin biosynthesis, we also analyzed the transcripts abundance of genes encoding for components of the IAA transport and transduction pathways in *Arabidopsis*, including *AUX1*, *IAA3/SHY2*, *PIN1*, *PIN2* [43]. Most of these genes remained unchanged in shoots and roots, except for *IAA3* which was slightly down-regulated in the shoot 48 h after inoculation (Figure 8). These results suggest that PS01 did not up-regulate IAA transport and transduction genes at transcriptional level.

To investigate whether PS01 altered the auxin distribution in root tissues, DR5::GUS reporter transgenic line was used. The micrographs indicated that the GUS staining was stronger and expanded in larger zones in roots of PS01-inoculated transgenic plants compared to the untreated roots (Figure 9). However, the GUS activity pattern may only reflect changes in auxin distribution within each root part without any changes in total root auxin content.

Asymmetric auxin distribution (auxin gradient) is the result of a polar auxin transport in plants. To evaluate the role of auxin transport on the alteration of RSA, an inhibitor of auxin polar transport, NPA was used. NPA inhibits root elongation and differential growth as well as lateral root development. Our results showed that the LRs formation was completely blocked in the presence of 1µM NPA. Furthermore, the PS01-induced LRs formation was affected in the presence of NPA. In PS01-inoculated seedlings, the number of LRs in the absence of NPA was 8.5 while that of NPA-treated seedlings was 6.5 (Figure 10a). In the presence of 5µM NPA, the formation of LR in the inoculated seedlings could not be observed even after 9 days of co-cultivation. This suggests that functional auxin efflux machinery is partially required for LR formation induced by PS01. In contrast, RH formation induced by PS01 showed no decrease upon NPA treatment (Figure 10b,c). To further evaluate the role of auxin transport in PS01-induced RH development, we evaluated the PS01-induced LR emergence in the auxin influx mutant aux1-7 and the auxin efflux carrier eir1-1 (Figure 10d,e). The ability of PS01 to promote RH development is retained in these mutants, suggesting the role of an independent auxin signaling pathway.

### 3.9. Potential of PS01 When Applied on Zea Mays

The *Pseudomonas* strain PS01 was isolated from maize rhizosphere; therefore, we also test the effects of this bacterial strain on the seed germination and the growth of maize plants under nutrients deficient condition were analyzed. The results showed that after 2 days of incubation, a higher rate of germination was observed in bacteria coated seeds (51.5%), compared to uncoated seeds (31.3%) (*n* = 100) (Appendix A). Previously, we reported the improvement of *Arabidopsis* germination rate in salt stress conditions by PS01 [45]. Here, we showed that after 30 days of growth in green-house conditions, inoculation of bacteria resulted in an increase in shoot and root dry weight by 1.4 fold and 2.2 fold, respectively (Figure 11a,b). A slight increase of the photosystem II efficiency (Fv/Fm) was also detected when maize plants were inoculated with PS01 (Figure 11c). In order to test root colonization by bacteria, the density of *Pseudomonas spp*. extracted from roots was assessed. There was a difference in *Pseudomonas* spp. population inn inoculated roots (6.7 × 10^4^ g^−1^ of fresh roots) compare to non-inoculated roots (9.7 × 10^1^ g^−1^ of fresh roots) (Appendix A). These results indicate that PS01 can improve germination, photosynthetic efficiency and growth in maize plants.

## 4. Discussion

*Pseudomonas* species are widely distributed in soil and plant roots. Indeed, *Pseudomonas* can be isolated successfully from almost every kind of agriculture soil or plant root. However, the inoculated bacteria sometimes cannot survive or perform in the same way in other soil and climatic conditions, because they must compete with the better-adapted indigenous microflora [32,55,56]. Therefore, it is important to isolate and identify region-specific microbial strains, which is capable of enhancing the crop yield of species adapted to specific ecological and environmental conditions. Currently, there are too few reports on PGP *Pseudomonas* isolated and characterized in the Mekong Delta [35,45]. In this study, a total of 20 strains were isolated from the rhizospheres of various plants and subjected to advance plant growth screening. Among these isolates, the PS01 strain significantly increased the shoot and root fresh weight of *Arabidopsis* seedlings.

Prior to application of microorganisms for agronomic purposes as a bio-inoculant, scenario safety testing and risk assessment evaluation was required. For instance, *P. aeruginosa* is an opportunistic human pathogen, whereas other species of *Pseudomonas* are non-pathogenic when used in agriculture [22,57]. In this context, potential strains must be taxonomically characterized at species and strain level using several approaches including whole genome sequencing and classified into the risk-group/ biosafety level (BSL) of the organism [57]. In this study, a draft whole-genome sequence of PS01 was established and showed that PS01 belongs to the *Pseudomonas putida* subclade. Interestingly, *P. putida* is classified in the risk group-1/BSL-1 class, which has been reported to have relative low harmful effects on human and the environment [58]. In few cases, low infectivity and virulence of *P. putida* strains have been reported in humans, however, these are rare and mostly reported in immuno-compromised individuals [58]. Therefore, considering the plant growth promoting characteristic and the inexistence of any harm to human health, PS01 can be considered as a promising strain for agronomic application. However, the presence of pathogenicity or virulence related genes in the PS01 genome should be further investigated in the future.

To understand and quantify the impact of PGPRs on roots and the whole plant, a common strategy was to inoculate roots with a PGPR In vitro and quantify effects on plant development. This methodology has shown that a large number of PGPRs may modify the RSA as well as root functionality, leading to the increase in uptake of nutrients and water and improve the overall growth of the whole plant [47]. A common pattern of root architecture adaptation to PGPR, including the reduction of the primary roots growth, was enhanced lateral root branching and development of root hairs was also documented [3,5,52,59]. In this study, PS01 altered the *Arabidopsis* RSA by inhibiting primary root elongation and promoting lateral root and root hair formation. These results are in agreement with the findings of earlier studies on *P. fluorescens* WCS417, *P. fluorescens* WCS374, *P. putida* WCS358, *S. marcescens* and *B. megaterium* [3,59,60]. In a previous report, more branching and a shorter root system was observed after inoculation with 25 of 40 *Pseudomonas* isolates [61].

Changes in RSA by PGPRs mainly resulted from their ability to interfere with the plants hormonal balance, implicating the production of phytohormones and other signals [47]. IAA is a well-characterized phytohormone produced by a large number of PGPRs [47]. Dose dependent effects of auxin are documented [12,62,63], wherein low concentrations of exogenous IAA stimulate primary root elongation while high IAA levels inhibit primary root growth, increase the formation of LRs and stimulate root hair development. In this study, the alteration of root development by PS01 was not blocked by the SCF^TIR1/AFB^ auxin receptor inhibitor (auxinole) and was not affected in the auxin mutants axr4-1, eir1-1, and aux1-7. The amount of auxin produced by PGPRs was directly correlated to the plant growth promotion and root development in a dose dependent manner. Interestingly, the alteration of root system architecture in response to *P. fluorescens* WCS417 and *B. megaterium* are similar to PS01 despite the fact that these PGPRs cannot produce IAA [3,60]. In another report where IAA production capacity of the *P. brassicacearum* STM196 strain was compared with the high-IAA-producing *A. brasilense* sp245 strain and its low-IAA-producing *ipd*C mutant. STM196 was found to not release significant amounts of auxin in the rhizosphere [43].

Some bacterial compounds other than auxin can also alter RSA in the same way as PS01 [47]. The bacterial signaling molecules, N-acyl-homoserine lactones (AHLs), confers quorum sensing, inhibits primary root growth, stimulates LR formation and promotes RH development [64,65]. AHL production was reported in 40% of *Pseudomonas* spp. colonizing plant roots including *P. chlororaphis*; *P. fluorescens* and *P. putida* [66]. Many PGPRs can produce the secondary metabolites that modulate the plant auxin interfering with the plant auxin pathway, such as 2,4-diacetyl phloroglucinol (DAPG), and nitric oxide. *A. brasilense* can produce NO during root colonization [47,64,67,68,69]. In fact, NO is involved in the lateral root formation controlled by auxin signaling pathway [67,69]. Similarly, DAPG can modify RSA through an auxin-dependent signaling pathway [64]. Indeed, exogenous DAPG inhibited primary root growth and stimulated LRs production in tomato seedlings at a concentration of around 10 μM [47]. VOCs produced by *Pseudomonas* WCS417 are not involved in WCS417-induced inhibition of primary root elongation but play an important role in promotion of LR formation [3]. Recently, an increased root branching capacity correlated with the production of cyclodipeptides by *P. putida* and *P. fluorescens* was reported [4].

Although it remains unknown whether auxin is a direct or in direct signalling molecule in alteration of RSA, most research groups suggested that auxin transport and signaling in plants may play an important role. Both primary-root elongation and lateral-root formation require auxin transport and an auxin gradient in the primary root tip [70]. A change in auxin distribution in response to PS01 inoculation was observed using the *Arabidopsis DR5*::*GUS* reporter line. This result is in agreement with the findings of earlier studies on *P. brassicacearum* STM196 and *Pseudomonas* WCS417 [3,43]. The stimulation effects of LR after inoculation may be mediated by auxin redistribution. Indeed, VOCs produced by *B. subtilis* GB03 can promote the growth of *Arabidopsis seedlings* by regulating auxin homeostasis in GB03-exposed roots as revealed by the DR5::GUS auxin-responsive *Arabidopsis* line [71]. This study also showed that PS01 can modify RSA in *Arabidopsis* via VOCs, which could act in an auxin-dependent or auxin-independent pathway. More than one secreted metabolite from strain PS01 may be involved in alteration of RSA. Therefore, screening for candidate strains based on a particular plant growth promoting trait such as IAA production can lead to overlooking many promising strains.

In our study, the potential of the *Pseudomonas* PS01 strain as phytostimulator was demonstrated. *Pseudomonas* PS01 improved seed germination and promoted vegetative growth in maize. These results reinforced the idea that PS01 could be used for the formulation of a new bio inoculant. Previously, we reported the ability of the PS01 strain in inducing a salt tolerance response in *Arabidopsis thaliana* [45]. However, successful application of this bacterium in soil conditions still requires further studies. The crucial factors in the success of *Pseudomonas* when applied in practical agriculture are related to their ability to colonize the rhizosphere and persistence throughout the growing season. Because they can exist in the natural habitat of the rhizospheres, they could colonize root surface when reintroduced through inoculation. Furthermore, the ability of PS01 to colonize and survive in rhizosphere should be assessed. Currently, the whole genome of PS01 had been sequenced, providing maximum of phylogenetic information for precise identification. Therefore, investigation of safety assessments such as presence of pathogenicity or virulence related genes may be also determined. In the future, it will be critical to conduct a more safety studies of Pseudomonas PS01 before biotechnological applications of this strain can be envisaged. Collectively, our results suggest that Pseudomonas PS01 can be developed for field application as an alternative to chemical fertilizers. Prediction and annotation gene sequences have been carried out to provide an in-depth view of the PGP characteristics of this strain, highlighting its potential use as a bio inoculant in agriculture.

## Figures and Tables

**Figure 1 microorganisms-08-00471-f001:**
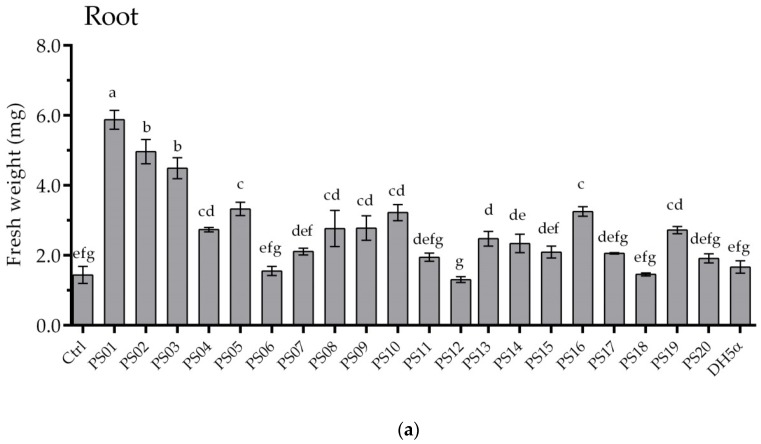
Screening of effective plant growth-promoting rhizobacteria (PGPR) strains using plant growth assay. Root (**a**) and shoot (**b**) fresh weight of *Arabidopsis* seedlings measured after 12 days of co-cultivation. *Pseudomonas* isolates were code as PS01–PS20. *E.coli* DH5 strain was used as bacterial control. The bars represent mean fresh weights ± SD (*n* = 15). Experiments were repeated at least 3 times with similar results. Different letters indicate statistically significant differences (Tukey’s HSD test; *p* <0.05).

**Figure 2 microorganisms-08-00471-f002:**
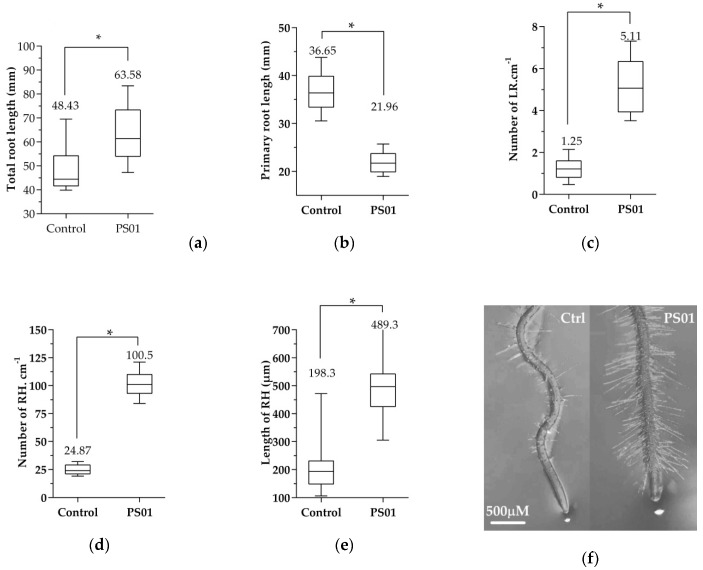
Effects of PS01 on root system architecture (RSA) of *Arabidopsis*. (**a**) Total root length was measured as total length of primary root and lateral roots (LR) (*n* = 15); (**b**) Primary root length (*n* = 15); (**c**) Density of lateral root was calculated as the number of LR divided by primary root length (*n* = 15); (**d**) Root hair (RH) density expressed as the average RH number ± SD in the root segment located 0.5 cm above the root tip (*n* =15); (**e**) Average RH length ± SD (*n* = 75); (**f**) Root tips of the non-inoculated (left) and inoculated (right) seedlings. Data are depicted as boxplots representing the range of values, the exclusive median. Asterisks indicate significant differences (Student’s *t* test; *p* < 0.05).

**Figure 3 microorganisms-08-00471-f003:**
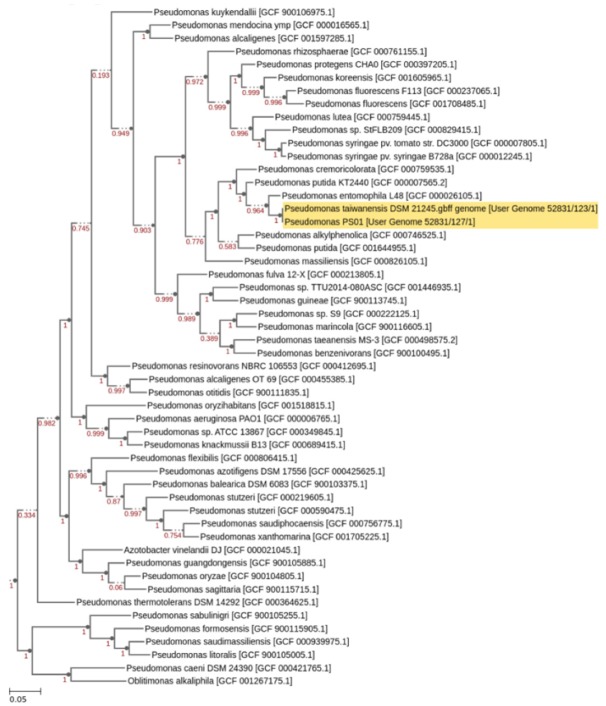
Taxonomic assignment of strain PS01. Phylogenetic tree was constructed by combining the genome of PS01 and *P. taiwanensis* DSM 21245 with a set of closely related genomes selected from all public KBase genomes using Insert Genome Into Species Tree 2.1.10 tool (http://kbase.us). Relatedness is determined by alignment similarity to a select subset of Clusters of Orthologous Groups domains. Next, a phylogenetic tree is reconstructed using FastTree (Version 2.1.10).

**Figure 4 microorganisms-08-00471-f004:**
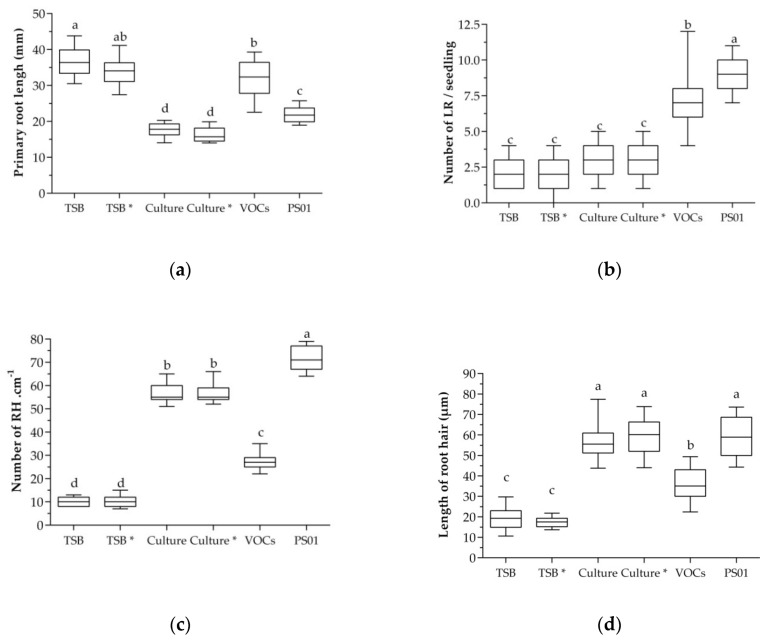
Different secreted metabolites produced by PS01 implicated in RSA of *Arabidopsis*. Effect of VOCs blend and diffusible compounds on (**a**) primary root elongation; (**b**) LRs formation; (**c**) RHs density and (**d**) RHs length were assessed. Cell-free culture supernatant of PS01 growing on TSB medium supplemented with (Culture*) or without tryptophan (Culture) was prepared by filtration. Then, 20 µL of filtrate was applied on a paper disk which was then placed in agar medium. TSB and TSB* (TSB supplemented with Trp) were used as control. VOCs treatment was performed by applying 20 µL of bacterial suspension (10^6^ CFU/mL) on a paper disk which was placed in the left side of the split plate. VOCs emitted by bacteria will expose to the seedlings at the left side. Data are depicted as boxplots representing the range of values, the exclusive median. Different letters indicate statistically significant differences (Tukey’s HSD test; *p* < 0.05).

**Figure 5 microorganisms-08-00471-f005:**
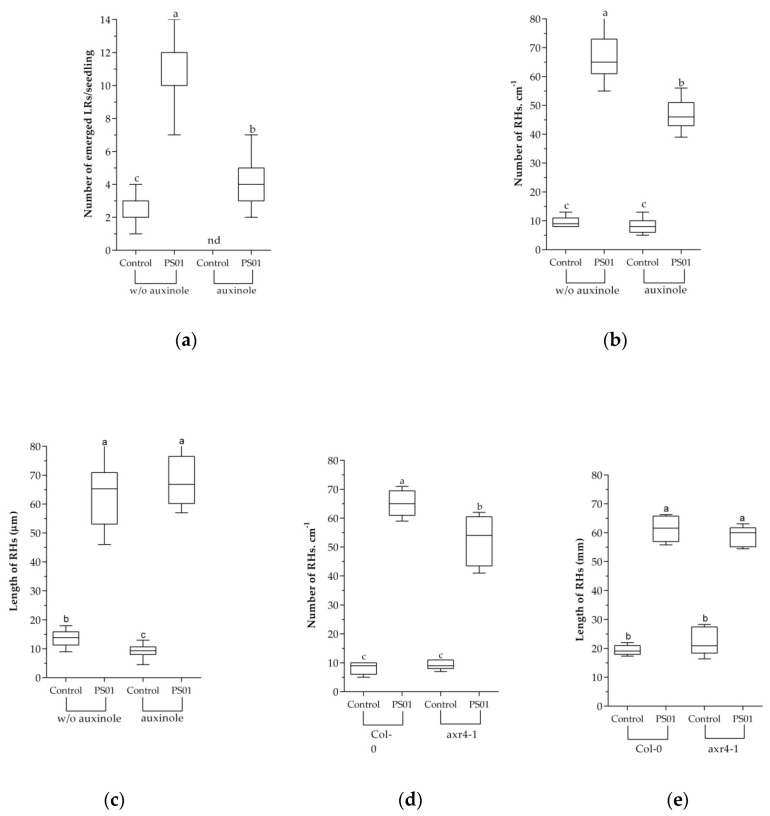
Role of bacterial auxin in PS01-mediated RSA alteration. Effects of the SCF^TIR1/AFB^) (SKP-Cullin-F box (SCF), Transport Inhibitor Resistant1/Auxin signaling F-box (TIR1/AFB)) auxin receptor inhibitor (auxinole) on (**a**) PS01-induced LR formation; (**b**) RH density; and (**c**) RH length in *Arabidopsis*. Assessment of (**d**) RH density and (**e**) RH length under control and PS01-induced conditions in wild-type roots (Col-0) and roots of auxin response mutant (axr4-1) after 9 d of co-cultivation. Data are depicted as boxplots representing the range of values, the exclusive median. n.d, not detected.

**Figure 6 microorganisms-08-00471-f006:**
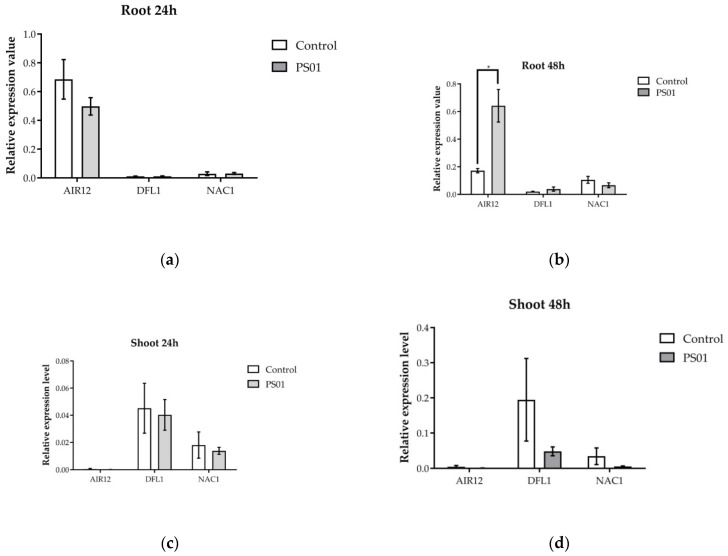
Effect of inoculation with PS01 on the expression levels of genes involved in auxin-dependent lateral root development regulation in *Arabidopsis* (*AIR12*, *DFL1*, *NAC1*) (Appendix A). RT-PCR analyses of gene expression have been performed separately on root (**a**,**b**) and shoot (**c**,**d**) at 24 and 48 h after inoculation. The bars presented are the mean expression levels ± SD (*n* = 3). Gene expression was normalized to Actin (At3 g18780) expression level. Asterisks indicate significant differences (Student’s *t* test; *p* < 0.05).

**Figure 7 microorganisms-08-00471-f007:**
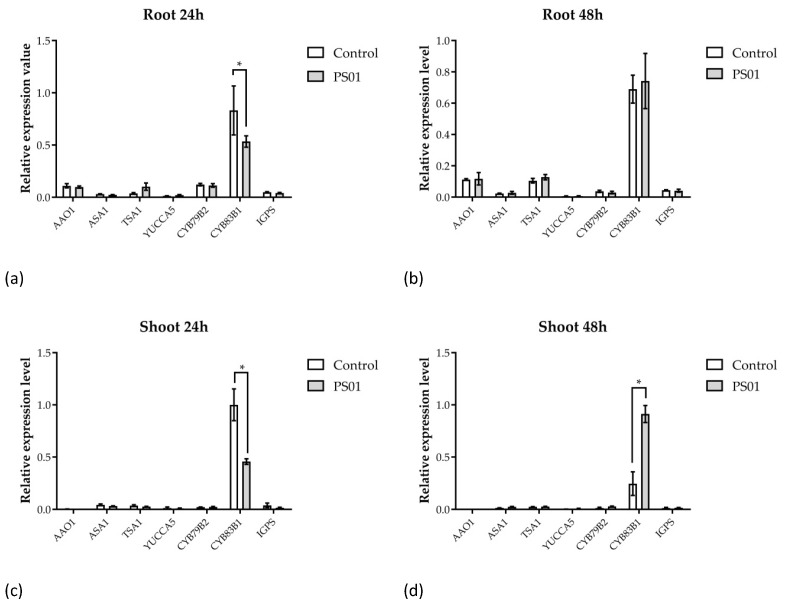
Effect of inoculation with PS01 on the expression levels of genes involved in tryptophan and auxin biosynthesis in *Arabidopsis*. RT-qPCR analyses of gene expression have been performed separately on root (**a**,**b**) and shoot (**c**,**d**) at 24 and 48 h after inoculation, as indicated. The bars presented are the mean expression levels ± SD (*n* = 3). Gene expression was normalized to Actin (At3 g18780) expression level. Asterisks indicate significant differences (Student’s *t* test; *p* < 0.05).

**Figure 8 microorganisms-08-00471-f008:**
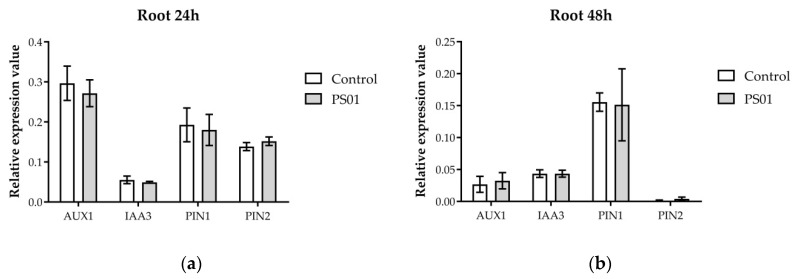
Effect of inoculation with PS01 on the expression levels of genes involved in auxin transport and transduction in Arabidopsis (AUX1, IAA3, PIN1, PIN2) (Appendix A). RT-qPCR analyses of gene expression have been performed separately on root (**a**,**b**) and shoot (**c**,**d**) at 24 and 48 h after inoculation, as indicated. The bars presented are the mean expression levels ± SD (*n* = 3). Gene expression was normalized to Actin (At3 g18780) expression level. Asterisks indicate significant differences (Student’s *t* test; *p* < 0.05).

**Figure 9 microorganisms-08-00471-f009:**
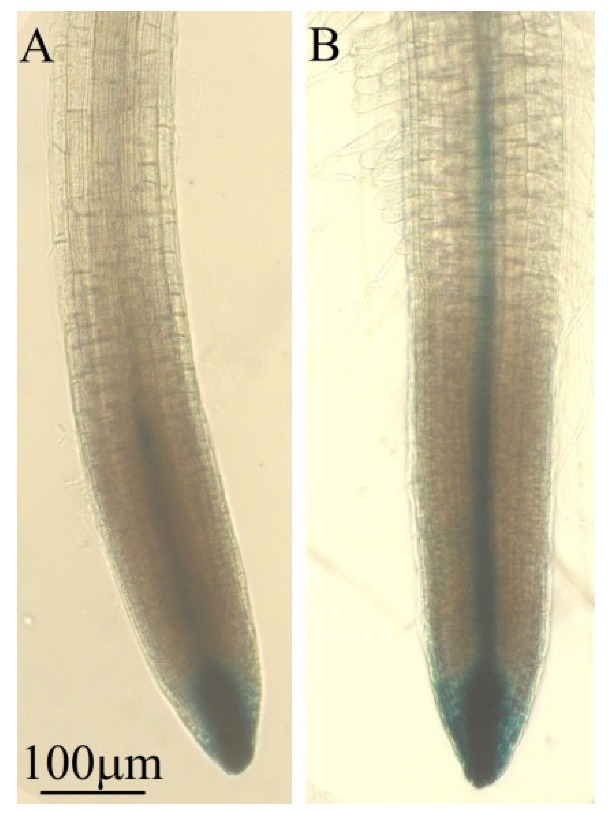
Effects of PS01 on auxin distribution in the *Arabidopsis* root. Images of DR5::GUS expression in mock treated root (**A**) and inoculated root (**B**) after 7 d of co-cultivation. Three replicates were performed with a total seedling population of 15; representative images are shown.

**Figure 10 microorganisms-08-00471-f010:**
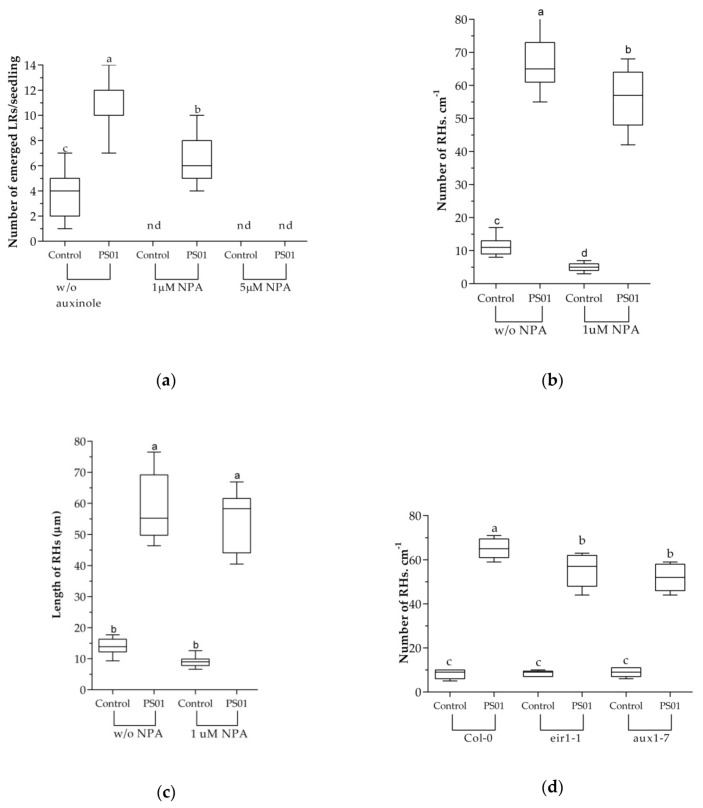
Influence of auxin transport on PS01-mediated RSA alteration in *Arabidopsis*. Effects of the polar auxin transport inhibitor (NPA) on (**a**) PS01-induced LR formation; (**b**) RH density; and (**c**) RH length. Assessment of (**d**) RH density and (**e**) RH length under control and PS01-induced conditions in wild-type roots (Col-0) and roots of auxin transport mutants (*eir1-1* and *aux1-7*) after 8 d of co-cultivation. Data are depicted as boxplots representing the range of values, the exclusive median. n.d, not detected.

**Figure 11 microorganisms-08-00471-f011:**
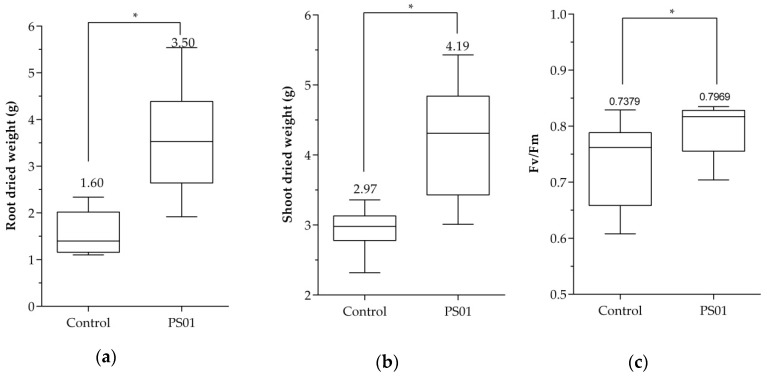
Effect of PS01 on growth of maize plant. PS01 improved (**a**) root dry weight; (**b**) shoot dry weight and (**c**) quantum efficiency of photosystem II in maize plants after 30 days. Data are depicted as boxplots representing the range of values, the exclusive median. Asterisks indicate significant differences (Student’s *t* test; *p* < 0.05).

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
