# Peer review of "Pseudomonas PS01 Isolated from Maize Rhizosphere Alters Root System Architecture and Promotes Plant Growth"

_microorganisms, 2020, doi:10.3390/microorganisms8040471_

Round 1

Reviewer 1 Report

Authors have investigated Pseudomonas PS01 for its potential role in improving the plant growth  using Arabidopsis and Corn plants. Overall the study is good and is of high relevance for exploring new bioinoculants for crop improvement.

Though may not be absolutely necessary for the paper but few more controls would be helpful to make the investigation more thorough.

Detection of Auxin and IAA concentration instead of just relying of the gene expression, TLC or other methods could be used for the same.

Explain in detail how authors isolated the VOCs, as the methods used could influence the outcome.

Could use a known strain for better controls and comparisons, something closes in there phylogenetic analysis could be used to understand specific benefits of using PS01 strain.

Author Response

Dear Reviewer,

We thank the reviewer for the positive comments and constructive suggestions. The major points raised by the reviewer are explained and addressed as follow.

  1. Detection of Auxin and IAA concentration instead of just relying of the gene expression, TLC or other methods could be used for the same.

Response 1: We agree that it is critical to detect or measure auxin and IAA concentration of PS01 inoculated roots and non-inoculated roots. We have tried to extract total auxin from roots of 10 days old seedlings and perform TLC. However, we could not detect auxin in TLC plate. Due to several limitations from our side including infrastructure, financial support and time constrain, we were unable to perform the HPLC or other methods for the current manuscript. We will try to optimize the IAA extraction and collaborate to Central Laboratory for Analysis (University of Science - Vietnam National University HCMC) to perform HPLC for measurement of IAA in the future.

  1. Explain in detail how authors isolated the VOCs, as the methods used could influence the outcome.

Response 2: We employed a split-plate assay in which bacteria and plants are separated into two parts by a septum that only allows bacterial VOCs to reach the seedlings.  The details of the experiment are included in the material & methods section at page 4 line 157 to 159 and in the figure 4 legend at page 11 line 365 to 367. In the supplementary figure 2S, we applied PS01 at right side and the VOCs will be produced and affected to the seedlings at the left side.

  1. Could use a known strain for better controls and comparisons, something closes in there phylogenetic analysis could be used to understand specific benefits of using PS01 strain.

Response 3: We agree that it will be better if we can use a known Pseudomonas strains to compare with our strain. We tried to contact different groups that are working with Pseudomonas to request their strains. Unfortunately, we did not receive any strains. We hope we will have a known Pseudomonas and perform field experiment to compare with PS01 in the future.

We thank you for your insightful comments.

Sincerely yours,

Minh Hoang

Reviewer 2 Report

The manuscript "Evaluation the plant growth-promoting effects and root system architecture alteration of Pseudomonas PS01 isolated from rhizosphere of maize grown on Mekong Delta" by Chu et al. presents a characterization of the interaction between Pseudomonas PS01 and the roots of Arabidopsis and maize, and also on seed germination. There is great interest in the field of plant growth-promoting bacteria, given the potential to contribute more sustainable methods to boost productivity. The study is well presented but the text needs to be revised by a native English speaker to meet the standards of the journal. Several grammar and syntax errors distracted me frequently while reading. One example that could be improved is the title, changing to "Pseudomonas PS01 isolated from maize rhizosphere promotes seed germination, plant growth and alter root system architecture".

Some minor points I would like the authors to address:

1) Line 291 and Lines 472-480: The authors claim that root hair promotion by PS01 is related to the combination of diffusible unknown compounds and VOCs. In the Discussion, it is mentioned some possible compounds but it is not clear what is known about compounds released by Pseudomonas. Please expand the Discussion if more is known specifically for Pseudomonas spp.

2) Figures 6, 7 and 8 show RT-PCR results on Arabidopsis genes. How come no RT-PCR was performed on PS01 genes, since several related to IAA synthesis were identified in the genome sequence? This seems to be an important source of data to back the claim that IAA secreted by the bacteria is not driving the change in root architecture. More genes involved with VOC or diffusible signal production could also have been analyzed.

3) Legend of Fig 11 needs revision.

4) Can the interaction of PS01 cells and roots and seeds be imaged? Maybe use a fluorescent strain to monitor their distribution on plant tissues? This would complement the data well. If authors cannot perform these experiments, please comment on what is known about their physical interaction.

Author Response

Dear Reviewer,

We are very grateful to the reviewer for the insightful comments and valuable suggestions. We have tried to address each of the reviewers concerns as described below.

The manuscript "Evaluation the plant growth-promoting effects and root system architecture alteration of Pseudomonas PS01 isolated from rhizosphere of maize grown on Mekong Delta" by Chu et al. presents a characterization of the interaction between Pseudomonas PS01 and the roots of Arabidopsis and maize, and also on seed germination. There is great interest in the field of plant growth-promoting bacteria, given the potential to contribute more sustainable methods to boost productivity. The study is well presented but the text needs to be revised by a native English speaker to meet the standards of the journal. Several grammar and syntax errors distracted me frequently while reading. One example that could be improved is the title, changing to "Pseudomonas PS01 isolated from maize rhizosphere promotes seed germination, plant growth and alter root system architecture".

Response: We extensively edited the language and style of the manuscript. We also corrected the grammar and spelling carefully. We modified the title as “Pseudomonas PS01 isolated from maize rhizosphere alters root system architecture and promotes plant growth”

Some minor points I would like the authors to address:

Point 1: Line 291 and Lines 472-480: The authors claim that root hair promotion by PS01 is related to the combination of diffusible unknown compounds and VOCs. In the Discussion, it is mentioned some possible compounds but it is not clear what is known about compounds released by Pseudomonas. Please expand the Discussion if more is known specifically for Pseudomonas spp.

Response1: We have modified the Discussion section, page 19, line 563 to 577 as follow: “Some bacterial compounds other than auxin can also alter RSA in the same way as PS01 [58]. The bacterial signaling molecules, N-acyl- homoserine lactones (AHLs), conferring quorum sensing, inhibited primary root growth, stimulated LR formation and promoted RH development [69][70]. AHL production was reported in 40% of Pseudomonas spp. colonizing plant roots including P. chlororaphis; P. fluorescens and P. putida [67 p.392]. Many PGPRs can produce the secondary metabolites that modulate the plant auxin interfering with the plant auxin pathway, such as 2,4-diacetyl phloroglucinol (DAPG), and nitric oxide (NO). A. brasilense can produce NO during root colonization [58,69,72–74]. In fact, NO is involved in the lateral root formation controlled by auxin signaling pathway [72,74]. Similarly, DAPG from P. fluorescens can modify RSA through an auxin-dependent signaling pathway [69]. Indeed, exogenous DAPG inhibited primary root growth and stimulated LR production in tomato seedlings at a concentration level of around 10 μM [58]. Role of VOCs produced by Pseudomonas WCS417 in alteration of RSA was reported. VOCs are not involved in WCS417-induced inhibition of primary root elongation but play an important role in promotion of LR formation [3]. Recently, an increased root branching capacity correlated with the production of cyclodipeptides by P. putida and P. fluorescens was reported [75].

Point 2: Figures 6, 7 and 8 show RT-PCR results on Arabidopsis genes. How come no RT-PCR was performed on PS01 genes, since several related to IAA synthesis were identified in the genome sequence? This seems to be an important source of data to back the claim that IAA secreted by the bacteria is not driving the change in root architecture. More genes involved with VOC or diffusible signal production could also have been analyzed.

Response 2: We are in full agreement with the reviewer. However, due to time constrain, we were unable to perform the suggested experiments for the current manuscript. Instead of performing RT-PCR, we used auxin inhibitors to assess the role of bacterial IAA, which was addressed in the Results section, Figure 5, page 12. We also assessed the role of VOCs through the split plate assay as presented in the Results section, Figure 4, page 11. Unfortunately, we cannot perform RT-qPCR for genes involved with VOCs or diffusible signal production. The reason is that there are too many candidate compounds, none of which has been exact compound has been identified, therefore, the workload will be too large to screen using the RT-PCR approach.

Point 3: Legend of Fig 11 needs revision.

Response 3: We have modified the Legend of Fig 11, page 18, line 496 to 500 as follow: “Effect of PS01 on growth of maize plant. PS01 improved (a) root dry weight; (b) shoot dry weight and (c) quantum efficiency of photosystem II in maize plants after 30 days. Data are depicted as boxplots representing the range of values, the exclusive median. Asterisks indicate significant differences (Student’s t test; p<0.05).”

Point 4: Can the interaction of PS01 cells and roots and seeds be imaged? Maybe use a fluorescent strain to monitor their distribution on plant tissues? This would complement the data well. If authors cannot perform these experiments, please comment on what is known about their physical interaction.

Response 4: We agree that it is critical to monitor the distribution of bacteria on the root system. However, due to several limitations from our side including infrastructure, financial support and time constrain, we were unable to develop a fluorescent tagged bacteria strain. Instead, we have tried to test the root colonization by bacteria and presented our data in the Results section, page 17, line 491 to 493 as follow: “In order to test root colonization by bacteria, the density of Pseudomonas spp. extracted from roots was assessed. There was a difference in Pseudomonas spp. population inn inoculated roots (6.7x104 g-1 of fresh roots) compare to non-inoculated roots (9.7x101 g-1 of fresh roots) (Fig S3).”

We thank you for your insightful comments.

Sincerely yours,

Minh Hoang

Reviewer 3 Report

This study by Chu and colleagues is an interesting and thorough exploration of the influence of PS01 on rhizosphere growth metrics. The amount of work and analyses which are presented are impressive. However, one criticism is that because there are so many results, not all are presented with enough information as to be reasonably interpreted. In many cases there is extensive interpretation of results in the results section, and presentation of new results in the discussion section. These sections will need to be rewritten or combined into a “results & discussion” section.

There are a huge number of figures in the manuscript, and although bar charts are useful in many cases, they can be unintentionally misleading. I would recommend box and whisker plots or, at a minimum, plotting the individual replicates as scatter points within each bar chart to show the true distribution and variability of individual replicates. For example, see this publication: https://bmcbiol.biomedcentral.com/articles/10.1186/s12915-015-0169-6 and https://journals.plos.org/plosbiology/article?id=10.1371/journal.pbio.1002128

Overall the writing is fine, although there are many instances where there is not subject-verb agreement or articles are missing from sentences. In many cases sentences are difficult to accurately interpret. A few instances are indicated below, however this manuscript would benefit from editing for clarity by an English language expert.

Other comments:

Line 16 (and throughout): refrain from using “next-generation sequencing”, as it’s unclear what exactly the next generation is beyond Illumina or MinION (etc), I would recommend using the newer convention of “high-throughput sequencing” instead.

Line 22: please clarify what is meant by “pharmacological approaches”, as there don’t appear to be any presented in this manuscript.

Line 31: “which and play” appears to be a typo

Line 35: spell out the entire species name the first time it is used, it would also be helpful for the reader to relate the scientific name to its common name (e.g. maize, mung bean, etc.)

Line 46: there appears to be a wayward comma in this sentence which is making it hard to interpret.

Line 52: the 2 in CO2 should be a subscript

Line 53: abbreviate VOCs

Line 59: this is an interesting point, are there any citations or examples for this kind of work?

Line 70: this sentence is potentially misleading, the case for linking climate change, chemical fertilizers, and soil degradation should be more explicitly linked to poor agricultural practices using appropriate citations.

Line 71: this sentence needs to be clarified; it is unclear how PGPR may be a promising strategy for food security (etc.) in response to climate change.

Line 77: this sentence is difficult to understand, please check.

Line 79: “…which was isolated from…”

Line 82: this sentence also need clarification; the phrasing makes it difficult to understand

Line 87: “… might be a promising strain…” might be more correct

Line 94: “grown for 18 h”

Line 108: what is meant by “uniform seedlings”? Is there an appropriate reference?

Line 113: how were the shoots and roots washed in ensure measurement of only the plants and not include any media?

Line 163: what are the other conditions for qPCR? What reaction efficiency was required for each run to be considered successful?

Line 169: was the cotton sterile?

Line 176: what were the replicate numbers for each of the treatments? How were plants washed prior to weighing? Was N=15 for each of the treatments?

Line 179: were the data tested for normality prior to using statistical tests with assumptions of normality?

Line 180: have these sequences been deposited into an online database?

Line 185: this sentence is very unclear and difficult to read accurately.

Line 192: it would be helpful for the reader if you clarified exactly what a clear zone on Pikovskaya’s agar indicates (either here or in the methods).

Line 194: how was this solubilization index calculated?

Line 199: reference figure 1 here so that the reader knows that this was statistically evaluated and the alpha level.

Line 203: was root disease observed on other plants? This seems like an out of place statement?

Line 204: more promising than what? Perhaps “the most promising” might be more clear phrasing?

Line 214: only in comparison to the control?

Line 228: figure 2f does not have a caption here.

Line 512: what do the asterixis indicate?

Line 247: is figure 3 referenced in the text?

Line 258 (and elsewhere): it is unclear why there are references in the results section, be careful of interpretation outside of the discussion section.

Line 279: this sentence is very confusing and should be rephrased.

Line 304: indicate in the figure caption what the VOCs are as well.

Line 323: “by contrary” is incorrect usage

Line 346: indicate in the figure caption what each of these genes on the X-axis are or refer explicitly to Table S1 (same comment for figure 8).

Line 357: this is also a confusing sentence

Line 393: “… could not be observed…”

Line 406: “crops” implies the investigators evaluated more than one type of crop plant.

Line 425: in many cases PGP-microorganisms have been shown to not be constrained by location or environmental condition (specifically endophytes do not suffer from this constraint), please be clearer with phrasing here.

Line 426: although the investigators here have authored a few relevant publications which should be referenced.

Line 430: It is not clear that this was evaluated in the presented work.

Lines 433 – 443: these are results and the first time they are presented, this should be in the results section.

Line 447: which minerals? The ones tested for herein?

Line 457: this statement requires references. What is meant by “best”?

Line 470: this sentence is repeated exactly from line 311

Line 481: why is it controversial? Include references or further discussion

Line 503: where and how is it available?

Line 503: there is not an in-depth discussion of the annotated gene sequences presented in this manuscript.

Author Response

Dear Reviewer,

We thank the reviewer for the valuable comments and constructive suggestions. The major and minor points raised by the reviewer are explained and addressed as follow.

  1. This study by Chu and colleagues is an interesting and thorough exploration of the influence of PS01 on rhizosphere growth metrics. The amount of work and analyses which are presented are impressive. However, one criticism is that because there are so many results, not all are presented with enough information as to be reasonably interpreted. In many cases there is extensive interpretation of results in the results section, and presentation of new results in the discussion section. These sections will need to be rewritten or combined into a “results & discussion” section.

Response 1: We agree that Results and Discussion sections are not well presented and properly interpreted. We made a major revision on “Results” and “Discussion”.

  1. There are a huge number of figures in the manuscript, and although bar charts are useful in many cases, they can be unintentionally misleading. I would recommend box and whisker plots or, at a minimum, plotting the individual replicates as scatter points within each bar chart to show the true distribution and variability of individual replicates. For example, see this publication: https://bmcbiol.biomedcentral.com/articles/10.1186/s12915-015-0169-6 and https://journals.plos.org/plosbiology/article?id=10.1371/journal.pbio.1002128

Response 2: We have replaced bar charts by box and whisker plots in the Figures 2, 4, 5, 10 and 11.

  1. Overall the writing is fine, although there are many instances where there is not subject-verb agreement or articles are missing from sentences. In many cases sentences are difficult to accurately interpret. A few instances are indicated below, however this manuscript would benefit from editing for clarity by an English language expert.

Response 3: We agree that the manuscript has English grammar mistakes. We extensively edited the language and style of the manuscript. We also corrected the grammar and spelling carefully.

Other comments:

Line 16 (and throughout): refrain from using “next-generation sequencing”, as it’s unclear what exactly the next generation is beyond Illumina or MinION (etc), I would recommend using the newer convention of “high-throughput sequencing” instead.

Response: We introduced “high-throughput sequencing” instead of “next-generation sequencing”

Line 22: please clarify what is meant by “pharmacological approaches”, as there don’t appear to be any presented in this manuscript.

Response: We used the chemical compounds such NPA and auxinole for auxin inhibitor analyses.

Line 31: “which and play” appears to be a typo

Response: We have deleted “which”

Line 35: spell out the entire species name the first time it is used, it would also be helpful for the reader to relate the scientific name to its common name (e.g. maize, mung bean, etc.)

Response: We agree with this suggestion. We have added the common name and the specie name as follow:

“…maize (Zea mays), soybean (Glycine max), wheat (Triticum), peanut (Arachis hypogaea) and mung bean (Vigna radiata)”

Line 46: there appears to be a wayward comma in this sentence which is making it hard to interpret.

Response:  We have modified the start of the paragraph as follow:

“Likewise, P. aeruginosa, P. putida and P. fluorescens can modulate…”

Line 52: the 2 in CO2 should be a subscript

Response: We have changed CO2 by CO2.

Line 53: abbreviate VOCs

Response: We have abbreviated it.

Line 59: this is an interesting point, are there any citations or examples for this kind of work?

Response: We have added the following citations:

  • Novik, G.; Savich, V.; Kiselev, E. An Insight Into Beneficial Pseudomonas bacteria. In Microbiology in Agriculture and Human Health; InTech, 2015.
  • Ferreira, C.M.H.; Soares, H.M.V.M.; Soares, E. V. Promising bacterial genera for agricultural practices: An insight on plant growth-promoting properties and microbial safety aspects. Sci. Total Environ. 2019, 682, 779–799.

Line 70: this sentence is potentially misleading, the case for linking climate change, chemical fertilizers, and soil degradation should be more explicitly linked to poor agricultural practices using appropriate citations.

Line 71: this sentence needs to be clarified; it is unclear how PGPR may be a promising strategy for food security (etc.) in response to climate change.

Response: We agree that the relation between PGPR and food security is not properly explained. Therefore, we have modified the sentence as follow:

“Application of PGPR as bio-inoculants, as a replacement for chemical fertilizer, therefore, is considered a sustainable agricultural practice. Moreover, it is a promising strategy for environmental safety and food security in response to climate changes”

Line 77: this sentence is difficult to understand, please check.

Response: We also consider that the sentence is unclear: Therefore, we have rephrased it as follow:

“However, the molecular mechanisms for enhancing plant growth and the genome sequences of native PGPRs have not been described yet.”

Line 79: “…which was isolated from…”

Response: We have modified the sentence following your suggestion.

Line 82: this sentence also need clarification; the phrasing makes it difficult to understand

Response: We have further clarified this point in our objective as following:

“Thereby, the present study aims to (i) screen and select a novel strain of Pseudomonas spp. which was isolated from rhizospheres of some crop plants grown on the Mekong Delta (ii) investigate effects of the isolate with respect to plant growth promotion and alteration of root system architecture (RSA), and (iii) analyze the relevant molecular mechanisms. The complete genome of a novel isolate, PS01 strain, was sequenced. We identified several gene clusters likely contributing to plant growth-promoting properties of PS01”

Line 87: “… might be a promising strain…” might be more correct

Response: We have modified the sentence as follow:

“….Pseudomonas PS01 could be commercialized as a bio-inoculant….”

Line 94: “grown for 18 h”

Response: We have modified the phrase following your suggestion.

Line 108: what is meant by “uniform seedlings”? Is there an appropriate reference?

Response: We have modified the sentence and added a citation as follow:

“….seedlings with similar size were transferred onto new (MS ½) solid medium plates as described previously [5]

Trinh, C.S.; Lee, H.; Lee, W.J.; Lee, S.J.; Chung, N.; Han, J.; Kim, J.; Hong, S.W.; Lee, H. Evaluation of the plant growth-promoting activity of Pseudomonas nitroreducens in Arabidopsis thaliana and Lactuca sativa. Plant Cell Rep. 2018, 37, 873–885.

Line 113: how were the shoots and roots washed in ensure measurement of only the plants and not include any media?

Response: To clarify this point, we have added the following sentence:

“Roots were washed carefully to get rid of medium prior to fresh biomass determination.”

Line 163: what are the other conditions for qPCR? What reaction efficiency was required for each run to be considered successful?

Response: We have added the RT-PCR conditions and PCR-product quality control as follow:

“The real-time PCR was performed on the Light Cycler 96 System (Roche). The reactions were incubated at 50°C for 2 min and 95°C for 10 min, followed by 40 cycles of 95°C for 15 s and 60°C for 1 min. The specificity of the real-time PCR amplification products was checked with the following dissociation protocol: heating at 95°C for 15 s, cooling at 60°C for 20 s, slowly heating up to 95°C within 20 min and a heating plateau at 95°C for 15s. Specific primer sets (Table S1) were reported previously [43]. To confirm the specificity, the size of PCR products were estimated by gel electrophoresis.”

Line 169: was the cotton sterile?

Response: Yes, the wet cotton was sterile. We have mentioned in the text as follow:

“…..sterilized wet cotton in…..”

Line 176: what were the replicate numbers for each of the treatments? How were plants washed prior to weighing? Was N=15 for each of the treatments?

Response: We explained more details in the materials and methods as following:

Soil adhering to the roots was carefully removed after rinsed in distillated water three times. The shoots and roots were separately harvested and dried in an oven for 2 days at 70°C. Then the shoot and root dried-weight of plants were recorded (N=15)."

Line 179: were the data tested for normality prior to using statistical tests with assumptions of normality?

Response: Yes, all the data met normal distribution and homogeneity of variance.

Line 180: have these sequences been deposited into an online database?

Response: We have not deposited into an online database

Line 185: this sentence is very unclear and difficult to read accurately.

Response: We also realize this sentence a bit confused. Therefore, we have modified the sentence as follow:

“In the previous studies, we found that Pseudomonas strain PS01 could enhance plant growth on saline conditions, whereas Pseudomonas strain P112 and P113 (coded as PS02 and PS03 in this study) enhanced plant growth on normal conditions.”

Line 192: it would be helpful for the reader if you clarified exactly what a clear zone on Pikovskaya’s agar indicates (either here or in the methods).

Response: We have clarified what the clear zone indicates, adding the following information:

“…a clear zone of calcium phosphate solubilisation on Pikovskaya’s agar plates…”

Line 194: how was this solubilization index calculated?

Response: We have added the following footnote in the Table 1:

(*) Phosphate solubilization index was derived by dividing the total diameter of the clear zone (colony+clear zone) with diameter of the colony.

Line 199: reference figure 1 here so that the reader knows that this was statistically evaluated and the alpha level.

Response: We have added the reference to Figure 1 in the suggested sentence.

Line 203: was root disease observed on other plants? This seems like an out of place statement?

Response: To avoid misunderstandings, we have decided eliminate the complete sentence.

Line 204: more promising than what? Perhaps “the most promising” might be more clear phrasing?

Response: We have modified the sentence following your suggestion.

Line 214: only in comparison to the control?

Response: Yes. Therefore, to clarify the meaning of the sentence, we have added the following phrase:

“…compared to control plants.”

Line 228: figure 2f does not have a caption here.

Response: We have added the following caption:

(f) Root tips of the non-inoculated (left) and inoculated (right) seedlings.

Line 512: what do the asterixis indicate?

Response: The meaning of the asterixis is described in the caption of Figure 4.

Line 247: is figure 3 referenced in the text?

Response: We have solved this mistake by adding Fig 3 reference in the following sentence:

“….PS01 is a member of Pseudomonas putida subclade and the closest relative is P. taiwanensis (Fig 3).”

Line 258 (and elsewhere): it is unclear why there are references in the results section, be careful of interpretation outside of the discussion section.

Response: We added some references in the Results section, because we want to remark some very important ideas (that are not specifically mentioned in the Introduction section) as starting point to describe the following experiments.

Line 279: this sentence is very confusing and should be rephrased.

Response: We have rephrased the sentence as follow:

“The results showed that primary root length was reduced approximately 50% and 40% in seedlings treated with cell-free culture supernatant of PS01 grown in medium with or without Trp (precursor of IAA), respectively, compared to the control plants (Figure 4a).”

Line 304: indicate in the figure caption what the VOCs are as well.

Response: We have added the next sentence:

“VOCs treatment was performed by applying 20 µL of bacterial suspension (106 CFU/ml) in a paper disk which was placed in the left side of the split plate. VOCs emissed by bacteria will expose to the seedlings at the left side.”

Line 323: “by contrary” is incorrect usage

Response: We have changed by “By contrast,…”

Line 346: indicate in the figure caption what each of these genes on the X-axis are or refer explicitly to Table S1 (same comment for figure 8).

Response: We have added the following phrases:

Figure 6: (AIR12, DFL1, NAC1) (Table S1)

Figure 8: (AUX1, IAA3, PIN1, PIN2) (Table S1)

Line 357: this is also a confusing sentence

Response: To clarify the meaning of the sentence, we have modified it as follow:

“Inoculation with PS01 down-regulated CYP83B1 expression in shoot after 24h of co-cultivation but induced an increase in its transcript abundance after 48h of co-cultivation (Fig 7).”

Line 393: “… could not be observed…”

Response: We have modified the phrase following your suggestion.

Line 406: “crops” implies the investigators evaluated more than one type of crop plant.

Response: We agree that this sentence is irrelevant in this section. Therefore, we have eliminated it.

Line 425: in many cases PGP-microorganisms have been shown to not be constrained by location or environmental condition (specifically endophytes do not suffer from this constraint), please be clearer with phrasing here.

Response: We agree with your suggestion and we added the following sentences to clarify this point.

“Indeed, Pseudomonas can be isolated successfully from almost every kind of agriculture soil or plant root. However, the inoculated bacteria sometimes cannot survive or perform in the same way in other soil and climatic conditions, because they must compete with the better-adapted indigenous microflora [32][53][54]. Therefore, it is important to isolate and identify region-specific microbial strains, which….”

Line 426: although the investigators here have authored a few relevant publications which should be referenced.

Response: We have added the following references:

Thanh, C.N.; Nhi, N.Y.; Diep, D.N.; Bao, T.T.H.; Minh, T.T.H.; Le, B. Van Evaluation of two Pseudomonas strains isolated from maize rhizosphere as plant growth promoting rhizobacteria. J Sci Techonol Dev. 2018, 2, 6–9.

Thanh, C.N.; Bao, T.H.T.; Le, V.B.; Minh, T.T.H. Plant growth-promoting rhizobacterium Pseudomonas PS01 induces salt tolerance in Arabidopsis thaliana. BMC Res. Notes 2019, 12, 11.

Line 430: It is not clear that this was evaluated in the presented work.

Response: We have not evaluated biosynthetic pathways that could produce any compound with potential toxic effects.

Lines 433 – 443: these are results and the first time they are presented, this should be in the results section.

Response: In fact, we propose that PS01 should be not harmful bacteria for humans, since a phylogenetic analysis showed that another member of the subclade, Pseudomonas putida, belongs to BSL group 1 (safe organism for humans). We have not annotated and analyzed the presence of pathogenic or virulence related genes in PS01 genome. Therefore, we keep this part at discussion

Line 447: which minerals? The ones tested for herein?

Response: We have written minerals, but the correct word is nutrients. Therefore, we have modified the sentence as following:

“….leading to the increase in uptake of nutrients and….”

Line 457: this statement requires references. What is meant by “best”?

Response: We have added references and changed “best” by “well” as following:

IAA is a well-characterized phytohormone produced by a large number of PGPRs [64][65]

Vacheron, J.; Desbrosses, G.; Bouffaud, M.L.; Touraine, B.; Moënne-Loccoz, Y.; Muller, D.; Legendre, L.; Wisniewski-Dyé, F.; Prigent-Combaret, C. Plant growth-promoting rhizobacteria and root system functioning. Front. Plant Sci. 2013, 4, 1–19.

Rhizobacteria, P.P.; Plants, M. Plant-Growth Promoting Rhizobacteria and Medicinal Plants; ISBN 9783319134000.

Line 470: this sentence is repeated exactly from line 311

Response: We have eliminated the sentence.

Line 481: why is it controversial? Include references or further discussion

Response: To clarify this point, we have eliminated “controversial” and modified a sentence as follow:

“Although it remains unknown whether auxin is a direct or in direct signalling molecule in alteration of RSA,, most research groups suggested that auxin transport and signaling in plants may play an important role.”

Line 503: where and how is it available?

Response: It was an error to mention that the genome sequence of PS01 is available. In fact, is not available. Therefore, we have changed the sentence by the following one:

“Currently, the whole genome of PS01 had been sequenced, providing maximum of phylogenetic information for precise identification. The presence of pathogenic or virulence related genes, therefore, may be also determined.”

Line 503: there is not an in-depth discussion of the annotated gene sequences presented in this manuscript.

Response: The physiological experiments and manuscript extension have not allowed us to discuss deeply about the annotated gene sequences. However, we have showed some annotated genes belonging to specific physiological pathways (Table S2). Moreover, we briefly discussed these genes in the Section 3.5.

Therefore, we have added as following:

“In the future, it will be critical to conduct a more safety studies of Pseudomonas PS01 before biotechnological applications of this strain can be envisaged. Collectively, our results suggest that Pseudomonas PS01 can be developed for field application as an alternative to chemical fertilizers.”

We thank you for your insightful comments.

Sincerely yours,

Minh Hoang

Round 2

Reviewer 2 Report

I congratulate the authors for this revised version. It has improved substantially and I will recommend acceptance for publication.

Author Response

Dear Reviewer,

We thank the reviewer for acceptance our manuscript for publication.

I congratulate the authors for this revised version. It has improved substantially and I will recommend acceptance for publication.

Response:  Thank you for your kind comment.

Best regards,

Minh Hoang

Reviewer 3 Report

This updated manuscript by Chu and colleagues has addressed many of my original concerns. There remains in the article typographic and grammatical errors throughout (some indicated below), these should be corrected prior to publication. 
Although not a requirement of this journal, I would highly recommend that the sequence data be deposited into an online, open-access database in the interested of transparency and scientific accountability.

Other comments:
Line 14: correct to “The full genome of PS01...”
Line 56: correct to “… at the species level...”
Line 81: missing a “.”
Line 81: editorial comment - recommend replacing “will be” with “could be” unless there are plans to apply this as a bio-inoculant?
Line 112: as a reader, I still cannot determine how the media was washed off. Using distilled or sterile water? Were the plants blotted dry prior to weighing? Or shaken? Or directly weighed (which could bias your data)?
Line 117: describe exactly how the seedlings were inoculated. Using the same methods described in section 2.4?
Line 143: define NGS (or perhaps you mean just “sequencing and annotation”
Line 187: indicate how the data were tested for normality.
Line 213: editorial comment - recommend “evaluate” in place of “dissect” as it has other connotations
Line 222: “fold” should be singular (here and elsewhere)
Line 228: results should be consistently written in past tense, please correct throughout
Line 276: are the authors implying that the VOCs are causing plant disease? Or do they mean “increasing plant resilience to systemic disease”?
line 318: “… analysis of the PS01…”
Line 462: this sentence is misleading as the authors did not test for harm to human health, rephrase to indicate that there were not genes which code for potential pathogenicity to humans (or similar).

Author Response

Dear Reviewer,

We are very grateful to the reviewer for the helpful comments and valuable suggestions. The minor points raised by the reviewer are explained and addressed as follow.

  1. This updated manuscript by Chu and colleagues has addressed many of my original concerns. There remains in the article typographic and grammatical errors throughout (some indicated below), these should be corrected prior to publication.

Response 1: We agree our manuscript still remains some typographic and grammatical errors. We have corrected the grammar and spelling carefully.

  1. Although not a requirement of this journal, I would highly recommend that the sequence data be deposited into an online, open-access database in the interested of transparency and scientific accountability.

Response 2: We are in full agreement with the reviewer on depositing PS01 genome sequences into an online, open-access database. Therefore, we have being submitted the sequences to GenBank.

Other comments:

Line 14: correct to “The full genome of PS01...”

Response: We have corrected to “The full genome of PS01...”

Line 56: correct to “… at the species level...”

Response: We have corrected to “... at the species level...”

Line 81: missing a “.”

Response: We have inserted a “.”

Line 81: editorial comment - recommend replacing “will be” with “could be” unless there are plans to apply this as a bio-inoculant?

Response: We have replaced “will be” by “could be”

Line 112: as a reader, I still cannot determine how the media was washed off. Using distilled or sterile water? Were the plants blotted dry prior to weighing? Or shaken? Or directly weighed (which could bias your data)?

Response:  To clarify this point, we have modified the sentence as following: “Media adhering to the roots were carefully removed using forceps. Then the roots were thoroughly rinsed with distilled water and blotted dry prior to fresh biomass determination.”

Line 117: describe exactly how the seedlings were inoculated. Using the same methods described in section 2.4?

Response: Here, we used the same method as described in section 2.4. Therefore, we have modified the sentence as follow: “To investigate the effect of bacteria on root development, the 4 days-old seedlings were inoculated with bacteria as described in section 2.4. Seven days after inoculation, seedling roots were photographed and analyzed using the ImageJ software...”

Line 143: define NGS (or perhaps you mean just “sequencing and annotation” 

Response: We have corrected to “Genome sequencing and annotation”

Line 187: indicate how the data were tested for normality.

Response: To further clarify this point, we have modified this as following: “...Data were first tested for normality with the D'Agostino & Pearson omnibus normality test and for homogeneity of variance with the Brown–Forsythe test. ANOVA followed by Tukey’s Honestly Significant Difference (HSD) test was performed for comparisons among all means and Student’s t test was performed for comparison of two means...”

Line 213: editorial comment - recommend “evaluate” in place of “dissect” as it has other connotations

Response: We have replaced “dissect” to “evaluate”

Line 222: “fold” should be singular (here and elsewhere)

Response: We have replaced “folds” to “fold” throughout.

Line 228: results should be consistently written in past tense, please correct throughout

Response: We have modified throughout the result section following your suggestion.

Line 276: are the authors implying that the VOCs are causing plant disease? Or do they mean “increasing plant resilience to systemic disease”?

Response: Here, we implied that the VOCs enhance disease resistance in plant. Therefore, to clarify this point, we modified this sentence as following:

“In addition to IAA, VOCs produced by bacteria may not only promote plant growth by modifying the root architecture system but also by playing an important role in inducing systemic resistance against plant pathogen.”

Line 318: “… analysis of the PS01…”

Response: We corrected to “... analysis of the PS01...”

Line 462: this sentence is misleading as the authors did not test for harm to human health, rephrase to indicate that there were not genes which code for potential pathogenicity to humans (or similar).

Response: We also realize this sentence a bit confused. Therefore, we edited the paragraph as following:

“...In this context, potential strains must be taxonomically characterized at species and strain level using several approaches including whole genome sequencing and classified into the risk-group/ biosafety level (BSL) of the organism [58]. In this study, a draft whole-genome sequence of PS01 was established and showed that PS01 belongs to the Pseudomonas putida subclade. Interestingly, P. putida is classified in the risk group-1/BSL-1 class, which has been reported to have relative low harmful effects on human and the environment [58]. In few cases, low infectivity and virulence of P. putida strains have been reported in humans, however, these are rare and mostly reported in immuno-compromised individuals [59]. Therefore, considering the plant growth promoting characteristic and the inexistence of any harm to human health, PS01 can be considered as a promising strain for agronomic application. However, the presence of pathogenicity or virulence related genes in the PS01 genome should be further investigated in the future.”

We thank you for your insightful comments.

Sincerely yours,

Minh Hoang